# AdaScale SGD: A Scale-Invariant Algorithm for Distributed Training

## Abstract

When using distributed training to speed up stochastic gradient descent, learning rates must adapt to new scales in order to maintain training effectiveness. Re-tuning these parameters is resource intensive, while fixed scaling rules often degrade model quality. We propose AdaScale SGD, a practical and principled algorithm that is approximately *scale invariant*. By continually adapting to the gradient's variance, AdaScale often trains at a wide range of scales with nearly identical results. We describe this invariance formally through AdaScale's convergence bounds. As the batch size increases, the bounds maintain final objective values, while smoothly transitioning away from linear speed-ups. In empirical comparisons, AdaScale trains well beyond the batch size limits of popular "linear learning rate scaling" rules. This includes large-scale training without model degradation for machine translation, image classification, object detection, and speech recognition tasks. The algorithm introduces negligible computational overhead and no tuning parameters, making AdaScale an attractive choice for large-scale training.

## 1 Introduction

Large datasets and large models underlie much of the recent success of machine learning. Training such models is time consuming, however, as stochastic gradient descent algorithms can require days or weeks to train effectively. Thus, procedures that speed up SGD are valuable. Faster training enables consideration of more data and models, which expands the capabilities of machine learning.

To speed up SGD, distributed systems can process thousands of training examples per iteration. But training at large scales also creates a major algorithmic challenge. Specifically, learning rates must adapt to each scale. Without choosing these training parameters carefully, scaled SGD frequently trains low-quality models, producing a waste of resources rather than a useful model.

To adapt learning rates, "fixed scaling rules" are standard but unreliable strategies. Goyal et al. (2017) popularized "linear learning rate scaling," which can work well, especially for computer vision tasks (Krizhevsky, 2014; Devarakonda et al., 2017; Jastrzębski et al., 2018; Smith et al., 2018; Lin et al., 2019). For other problems or larger scales, however, linear scaling often fails. This fact is well-known in theory (Yin et al., 2018; Jain et al., 2018; Ma et al., 2018) and in practice (Goyal et al., 2017). Other fixed scaling rules are also undependable. Golmant et al. (2018) test three rules—linear, root, and identity—and conclude that each one often degrades model quality. Shallue et al. (2019) compute near-optimal parameters for many tasks and scales, and the results do not align with any fixed rule. To ensure effective training, the authors recommend avoiding such rules and re-tuning parameters for each new scale—an inconvenient and resource-intensive solution.

We propose AdaScale SGD. A practical but principled algorithm, AdaScale more reliably scales training by adapting to the gradient's variance. Decreased gradient variance is the fundamental impact of large batch sizes. Thus, scaling provides little gain if the variance is already "small" at small scales. In such cases, AdaScale increases the learning rate conservatively, and large-scale training progresses similarly to the small-batch setting. For iterations with "large" gradient variance, AdaScale increases the learning rate aggressively, and the per-iteration progress dramatically increases.

AdaScale is approximately scale invariant, a quality that simplifies large-batch training. With no changes to learning rates or other inputs, AdaScale can train at many scales with similar results. This leads to two important innovations: (i) AdaScale improves the translation of training configurations between scales, which is useful for scaling up tasks or adapting to dynamic resource availability;

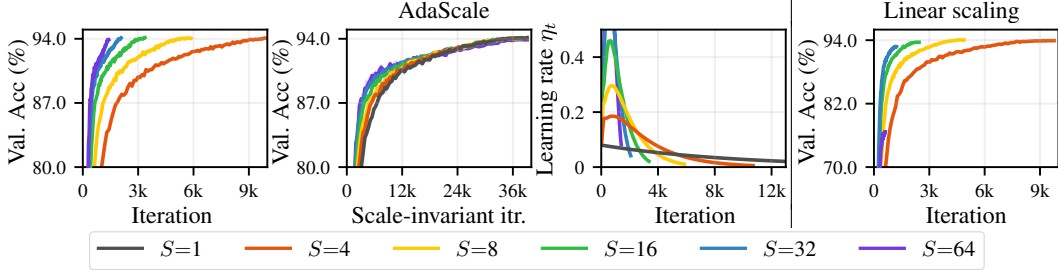

**Figure 1: Motivating results.** For `cifar10`, AdaScale preserves model quality for many scales $S$. When plotted in terms of scale-invariant iterations, training curves align closely. With AdaScale, "warm-up" behavior emerges from adapting a simple learning rate schedule (exponential decay) to scale $S$ (learning rate plot cropped to show behavior). Meanwhile, linear scaling (with warm-up heuristic) degrades model quality as $S$ increases.

and (ii) AdaScale works at scale with simple learning rate schedules, which eliminates the need for "warm-up" heuristics (Goyal et al., 2017). Qualitatively, AdaScale and warm-up have similar effects on learning rates, but with AdaScale, this behavior emerges from a principled and adaptive mechanism, not hand-tuned parameters.

We provide theoretical results that formalize this approximate scale invariance. Bounds for all scales converge to identical objective values. In contrast, the linear scaling rule requires fewer iterations but compromises model quality and training stability, causing divergence as the batch size increases.

We perform large-scale empirical evaluations on five training benchmarks. Tasks include image classification, machine translation, object detection, and speech recognition. The results align well with our theory, as AdaScale systematically preserves model quality across many scales. This includes training ImageNet with batch size 32k and Transformer with 262k max tokens per batch.

To provide context for our description of AdaScale, Figure 1 includes results from a simple scaling experiment using CIFAR-10 data. These results illustrate the concept of scale invariance, AdaScale's qualitative impact on learning rates, and a failure case for the linear scaling rule.

## 2 PROBLEM FORMULATION

We focus on quickly computing approximate solutions to the problem

$$\text{minimize}_{\mathbf{w} \in \mathbb{R}^d} \ F(\mathbf{w}), \quad \text{where} \quad F(\mathbf{w}) = \mathbb{E}_{\mathbf{x} \sim \mathcal{X}} \left[ f(\mathbf{w}, \mathbf{x}) \right] . \tag{P1}$$

Here $\mathbf{w}$ parameterizes a machine learning model, while $\mathcal{X}$ denotes a distribution over batches of training data. We assume that $F$ and $f$ are differentiable and that $\mathbb{E}_{\mathbf{x} \sim \mathcal{X}} \left[ \nabla_{\mathbf{w}} f(\mathbf{w}, \mathbf{x}) \right] = \nabla F(\mathbf{w})$.

Stochastic gradient descent is a popular algorithm for solving (P1). Let $\mathbf{w}_t$ denote the model parameters when iteration $t$ begins. During this iteration, SGD samples a batch $\mathbf{x}_t \sim \mathcal{X}$ and computes the gradient $\mathbf{g}_t \leftarrow \nabla_{\mathbf{w}} f(\mathbf{w}_t, \mathbf{x}_t)$. SGD then applies the update $\mathbf{w}_{t+1} \leftarrow \mathbf{w}_t - \eta_t \mathbf{g}_t$. Here $\eta_t$ is the *learning rate*. Given a schedule $\text{lr} : \mathbb{Z}_{\geq 0} \rightarrow \mathbb{R}_{>0}$, SGD defines $\eta_t = \text{lr}(t)$. For our experiments in §4, `lr` is an exponential decay or step decay function. SGD completes training after $T$ iterations.

To speed up training, practitioners often parallelize gradient computation across multiple devices. Algorithm 1 defines a scaled SGD algorithm. At scale $S$, the algorithm samples $S$ independent batches during each iteration. After computing the gradient for each batch in parallel, the algorithm applies the mean of these gradients (in place of $\mathbf{g}_t$) when updating model parameters.

But scaling training in this way creates a considerable algorithmic challenge. Each new scale requires a new learning rate schedule, which is inconvenient and resource intensive to obtain. To help address this challenge, we propose a scaled SGD algorithm that is approximately *scale invariant*.

**Definition 1.** *Let $\mathbf{w}_T$ denote the (possibly random) result of a scaled SGD algorithm. Fixing all algorithm inputs except scale $S$, the algorithm is **scale invariant** if $\mathbf{w}_T$ does not depend on $S$.*

A scale-invariant algorithm makes parallelizing training significantly easier. Such an algorithm can scale to any available amount of computational resources, and there is no need for parameter re-tuning, unreliable heuristics, or algorithmic expertise from users.

---

**Algorithm 1** Scaled SGD

   **function** Scaled_SGD$(S, \texttt{lr}, T, \mathcal{X}, f, \mathbf{w}_0)$
      **for** $t = 0, 1, 2, \ldots, T-1$ **do**
         $\bar{\mathbf{g}}_t \leftarrow$ compute_gradient$(\mathbf{w}_t, S, \mathcal{X}, f)$
         $\eta_t \leftarrow \texttt{lr}(t)$
         $\mathbf{w}_{t+1} \leftarrow \mathbf{w}_t - \eta_t \bar{\mathbf{g}}_t$
      **return** $\mathbf{w}_T$

---

   **function** compute_gradient$(\mathbf{w}_t, S, \mathcal{X}, f)$
      **in parallel for** $i = 1, \ldots, S$ **do**
         $\mathbf{x}^{(i)} \leftarrow$ sample_batch$(\mathcal{X})$
         $\mathbf{g}^{(i)} \leftarrow \nabla_{\mathbf{w}} f(\mathbf{w}_t, \mathbf{x}^{(i)})$
      **return** $\frac{1}{S} \sum_{i=1}^{S} \mathbf{g}^{(i)}$

---

**Algorithm 2** AdaScale SGD

   **function** AdaScale$(S, \texttt{lr}, T_{\text{SI}}, \mathcal{X}, f, \mathbf{w}_0)$
      **initialize** $\tau_0 \leftarrow 0; t \leftarrow 0$
      **while** $\tau_t < T_{\text{SI}}$ **do**
         $\bar{\mathbf{g}}_t \leftarrow$ compute_gradient$(\mathbf{w}_t, S, \mathcal{X}, f)$

         *# Compute gain $r_t \in [1, S]$ (see §3.3):*
         $r_t \leftarrow \dfrac{\mathbb{E}\left[\sigma^2(\mathbf{w}_t) + \|\nabla F(\mathbf{w}_t)\|^2\right]}{\mathbb{E}\left[\frac{1}{S}\sigma^2(\mathbf{w}_t) + \|\nabla F(\mathbf{w}_t)\|^2\right]}$

         $\eta_t \leftarrow r_t \cdot \texttt{lr}(\lfloor \tau_t \rfloor)$
         $\mathbf{w}_{t+1} \leftarrow \mathbf{w}_t - \eta_t \bar{\mathbf{g}}_t$
         $\tau_{t+1} \leftarrow \tau_t + r_t; t \leftarrow t + 1$
      **return** $\mathbf{w}_t$

---

## 3 ADASCALE SGD ALGORITHM

This section introduces our AdaScale algorithm. As motivation, we first consider the role of gradient variance in SGD. We later provide practical guidance for variance estimation and momentum tuning.

### 3.1 INTUITION: IDENTITY SCALING, LINEAR SCALING, AND GRADIENT VARIANCE

We now consider two fixed scaling rules, which influence the design of AdaScale. One of these rules is identity scaling, which keeps the training configuration constant for all scales:

**Definition 2.** *To apply the **identity scaling rule** to Algorithm 1, use the same* lr *and* $T$ *for all* $S$.

Note that this rule has little practical appeal, since it fails to reduce the number of training iterations. A second and more popular strategy is linear learning rate scaling:

**Definition 3.** *To apply the **linear learning rate scaling rule** to Algorithm 1, use* lr$(t) = S \cdot$lr$_{S1}(St)$ *and* $T = \lceil T_{S1}/S \rceil$, *where* lr$_{S1}$ *and* $T_{S1}$ *denote the learning rate schedule and total steps for* $S = 1$.

Conceptually, linear scaling treats SGD as a perfectly parallelizable algorithm. If true, applying gradients from $S$ batches in parallel achieves the same result as doing so in sequence.

For special cases of (P1), the identity and linear rules result in scale-invariant algorithms. To show this, we first define the variance quantities

$$\boldsymbol{\Sigma}(\mathbf{w}) = \text{cov}_{\mathbf{x} \sim \mathcal{X}}(\nabla_{\mathbf{w}} f(\mathbf{w}, \mathbf{x}), \nabla_{\mathbf{w}} f(\mathbf{w}, \mathbf{x})), \quad \text{and} \quad \sigma^2(\mathbf{w}) = \text{tr}(\boldsymbol{\Sigma}(\mathbf{w})).$$

In words, $\sigma^2(\mathbf{w})$ sums the variances of each entry in $\nabla_{\mathbf{w}} f(\mathbf{w}, \mathbf{x})$. By sampling batches independently, scaling fundamentally impacts SGD by reducing this variance. Given $\mathbf{w}_t$ in Algorithm 1, we have $\text{cov}(\bar{\mathbf{g}}_t, \bar{\mathbf{g}}_t) = \frac{1}{S}\boldsymbol{\Sigma}(\mathbf{w}_t)$ and $\mathbb{E}[\bar{\mathbf{g}}_t] = \nabla F(\mathbf{w}_t)$. Here, only the covariance depends on $S$.

Consider the special case of zero gradient variance. In this case, identity scaling performs ideally:

**Proposition 1** (Scale-invariant SGD for deterministic gradients). *If* $\sigma^2(\mathbf{w}) = 0$ *for all* $\mathbf{w} \in \mathbb{R}^d$, *then applying identity scaling to Algorithm 1 results in a scale-invariant algorithm.*

Although identity scaling does not speed up training, Proposition 1 is critical for framing the impact of large scales. If the gradient variance is "small," then we cannot expect large gains from increasing $S$—a larger scale has little effect on $\bar{\mathbf{g}}_t$. With "large" variance, however, the opposite is true:

**Proposition 2** (Scale-invariant SGD for extreme stochasticity). *Consider fixed covariance matrix* $\tilde{\boldsymbol{\Sigma}} \in \mathbb{S}_{++}^d$, *learning rate value* $\tilde{\eta} \in \mathbb{R}_{>0}$, *and training duration* $\tilde{T}$. *For a given* $\nu \in \mathbb{R}_{>0}$, *assume* $\nabla_{\mathbf{w}} f(\mathbf{w}, \mathbf{x}) \sim \mathcal{N}(\nabla F(\mathbf{w}), \nu \tilde{\boldsymbol{\Sigma}})$, *and apply linear scaling to Algorithm 1 with* lr$_{S1}(t) = \nu^{-1}\tilde{\eta}$ *and* $T_{S1} = \nu \tilde{T}$. *The resulting scaled SGD algorithm is scale-invariant in the limit* $\nu \to +\infty$.

In less formal terms, linear scaling leads to scale-invariance in the case of very large gradient variance (as well as small learning rates and many iterations, to compensate for this variance). Since increasing $S$ decreases variance, it is natural that scaling yields large speed-ups in this extreme case.

In practice, the gradient's variance is neither zero nor infinite, and both identity and linear scaling may perform poorly. Moreover, the gradient's variance does not remain constant throughout training. A scale-invariant algorithm, it seems, must continually adapt to the state of training.

## 3.2 ADASCALE DEFINITION

AdaScale, defined in Algorithm 2, adaptively interpolates between identity and linear scaling, based on the expectation of $\sigma^2(\mathbf{w}_t)$. During iteration $t$, AdaScale multiplies the learning rate by the "gain ratio" $r_t \in [1, S]$: $\eta_t = r_t \cdot \mathtt{lr}(\lfloor \tau_t \rfloor)$. Here $\tau_t$ is the "scale-invariant iteration," defined as $\tau_t = \sum_{t'=0}^{t-1} r_{t'}$. The idea is that iteration $t$ performs the equivalent of $r_t$ single-batch iterations, and $\tau_t$ accumulates this progress. AdaScale concludes when $\tau_t \geq T_{\text{SI}}$, where $T_{\text{SI}}$ is the total scale-invariant iterations. Since $r_t \in [1, S]$, AdaScale requires at least $\lceil T_{\text{SI}}/S \rceil$ and at most $T_{\text{SI}}$ iterations.

The identity and linear rules correspond to two special cases of AdaScale. If $r_t = 1$ for all $t$, the algorithm equates to SGD with identity scaling. Similarly, if $r_t = S$ for all $t$, we have linear scaling. Thus, to approximate scale-invariance, §3.1 suggests setting $r_t \approx 1$ when the gradient's variance is small and $r_t \approx S$ when this variance is large. AdaScale achieves this by defining

$$r_t = \mathbb{E}\left[\sigma^2(\mathbf{w}_t) + \|\nabla F(\mathbf{w}_t)\|^2\right] \Big/ \mathbb{E}\left[\tfrac{1}{S}\sigma^2(\mathbf{w}_t) + \|\nabla F(\mathbf{w}_t)\|^2\right].$$

The expectations here are with respect to the distribution of $\mathbf{w}_t$, and we must approximate $r_t$ in practice (see §3.3). This definition of $r_t$ ensures that as $S$ increases, $\mathbb{E}[\langle \mathbf{w}_{t+1} - \mathbf{w}_t, \nabla F(\mathbf{w}_t)\rangle]$ and $\mathbb{E}[\|\mathbf{w}_{t+1} - \mathbf{w}_t\|^2]$ increase multiplicatively by $r_t$. This leads to our scale-invariant bound in §5.

## 3.3 PRACTICAL CONSIDERATIONS

If $S = 1$ in AdaScale, then $r_t = 1$ for all iterations. For larger scales, $r_t$ depends on $\mathbb{E}\left[\sigma^2(\mathbf{w}_t)\right]$ and $\mathbb{E}\left[\|\nabla F(\mathbf{w}_t)\|^2\right]$, and a practical implementation must efficiently approximate these values. Fortunately, the per-batch gradients $\mathbf{g}_t^{(1)}, \ldots, \mathbf{g}_t^{(S)}$ and aggregated gradient $\bar{\mathbf{g}}_t$ are readily available in distributed SGD algorithms. This makes approximating $r_t$ straightforward. In particular, we define

$$\hat{\sigma}_t^2 = \tfrac{1}{S-1}\sum_{i=1}^{S} \|\mathbf{g}_t^{(i)}\|^2 - \tfrac{S}{S-1}\|\bar{\mathbf{g}}_t\|^2,$$

$$\text{and}\quad \hat{\mu}_t^2 = \|\bar{\mathbf{g}}_t\|^2 - \tfrac{1}{S}\hat{\sigma}_t^2.$$

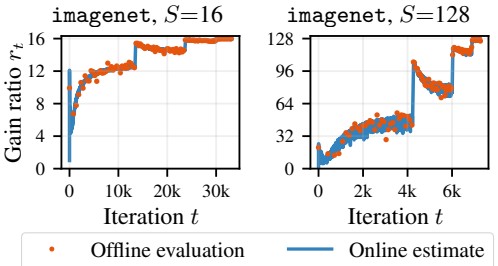

**Figure 2: Gain ratios**. Plots compare moving average $r_t$ estimates to values computed offline (using 1000 batches). The values align closely. Abrupt changes align with learning rate step changes.

Here $\hat{\sigma}_t^2$ and $\hat{\mu}_t^2$ are unbiased estimates of $\mathbb{E}\left[\sigma^2(\mathbf{w}_t)\right]$ and $\mathbb{E}\left[\|\nabla F(\mathbf{w}_t)\|^2\right]$. To ensure robustness to estimation variance, we estimate $r_t$ by plugging in moving averages $\bar{\sigma}_t^2$ and $\bar{\mu}_t^2$, which average $\hat{\sigma}_t^2$ and $\hat{\mu}_t^2$ over prior iterations. Our implementation uses exponential moving average parameter $\theta = \max\{1 - S/1000, 0\}$, where $\theta = 0$ results in no averaging. We find that AdaScale is robust to the choice of $\theta$, and we provide evidence of this in Appendix C. To initialize, we set $r_0 \leftarrow 1$, and for iterations $t < (1-\theta)^{-1}$, we define $\bar{\sigma}_t^2$ and $\bar{\mu}_t^2$ as the mean of past samples. Before averaging, we clip estimates so that $\hat{\sigma}_t^2 \geq 10^{-6}$ (to prevent division by zero) and $\hat{\mu}_t^2 \geq 0$ (to ensure $r_t \in [1, S]$).

To verify these estimators, Figure 2 compares moving average estimates to offline estimates using model checkpoints. These plots also provide examples of gain ratios for practical problems. We note that numerous prior works—for example, (Schaul et al., 2013; Kingma & Ba, 2015; McCandlish et al., 2018)—have relied on similar moving averages to estimate gradient moments.

One final practical consideration is the momentum parameter $\rho$ when using AdaScale with momentum-SGD. The performance of momentum-SGD depends less critically on $\rho$ than the learning rate (Shallue et al., 2019). For this reason, we find that AdaScale often performs well if $\rho$ remains constant across scales and iterations. This approach to momentum scaling has also succeeded in prior works involving the linear scaling rule (Goyal et al., 2017; Smith et al., 2018).

## 4 EMPIRICAL COMPARISONS

We evaluate AdaScale on five practical training benchmarks. We assess scale invariance by comparing training curves across scales. We assess impact on training times by comparing total iterations. We consider a variety of tasks, models (He et al., 2016a;b; Amodei et al., 2016; Vaswani et al., 2017; Redmon & Farhadi, 2018), and datasets (Deng et al., 2009; Krizhevsky, 2009; Everingham et al., 2010; Panayotov et al., 2015). Table 1 summarizes our training benchmarks. Due to space limitations, we provide additional implementation details in Appendix B.

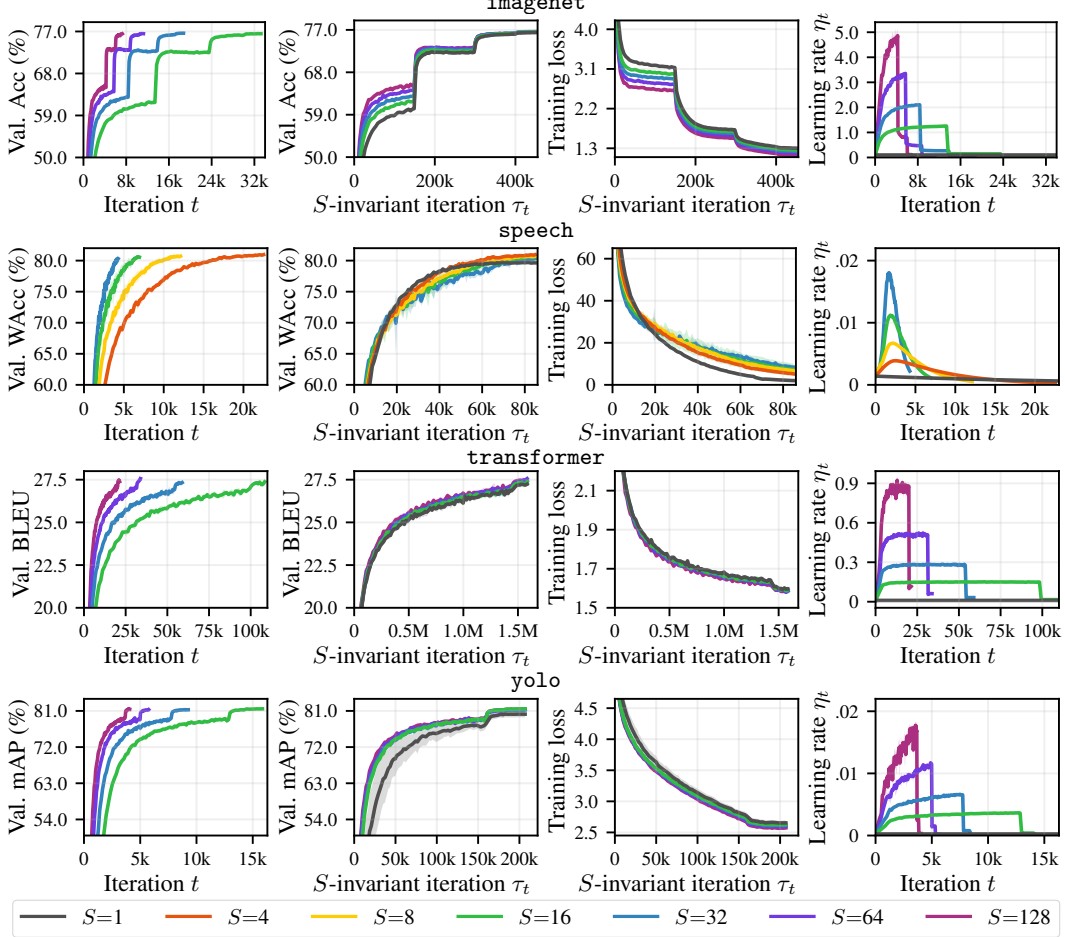

**Figure 3: AdaScale training curves.** For many scales and benchmarks, AdaScale trains quality models. Training curves align closely in terms of $\tau_t$. In all cases, $\eta_t$ warms up gradually at the start of training, even though all `lr` schedules are simple exponential or step decay functions (which are non-increasing in $t$).

For each benchmark, we use *one simple learning rate schedule*. Specifically, `lr` is an exponential decay function for `cifar10` and `speech`, and a step decay function otherwise. We use standard `lr` parameters for `imagenet` and `yolo`. Otherwise, we use tuned parameters that approximately maximize the validation metric (to our knowledge, there are no standard schedules for solving `speech` and `transformer` with momentum-SGD). We use momentum $\rho = 0.9$ except for `transformer`, in which case we use $\rho = 0.99$ for greater training stability.

Figure 3 (and Figure 1) contains AdaScale training curves for the benchmarks and many scales. Each curve plots the mean of five distributed training runs with varying random seeds. As $S$ increases, AdaScale trains for fewer iterations but consistently preserves model quality. Illustrating AdaScale's approximate scale invariance, the training curves align closely when plotted in terms of scale-invariant iterations.

**Table 1: Overview of training benchmarks.**

| Name | Task | Model | Dataset | Metric |
|------|------|-------|---------|--------|
| `cifar10` | Image classification | ResNet-18 (v2) | CIFAR-10 | Top-1 accuracy (%) |
| `imagenet` | Image classification | ResNet-50 (v1) | ImageNet | Top-1 accuracy (%) |
| `speech` | Speech recognition | Deep speech 2 | LibriSpeech | Word accuracy (%) |
| `transformer` | Machine translation | Transformer base | WMT-2014 | BLEU |
| `yolo` | Object detection | YOLOv3 | PASCAL VOC | mAP (%) |

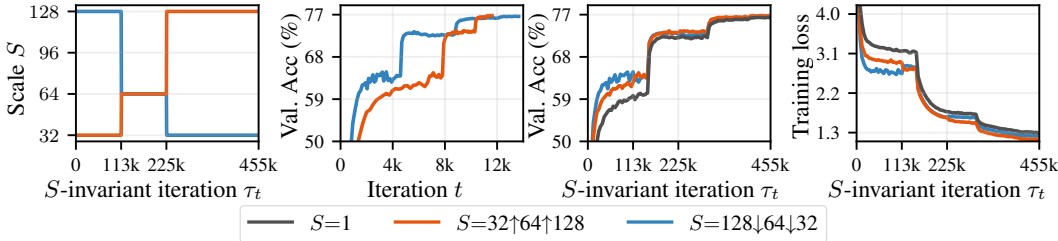

**Figure 4: Elastic AdaScaling.** For `imagenet`, AdaScale is approximately scale invariant, even if $S$ changes abruptly (at $\tau_t = 133k, 225k$). Unlike AdaScale, LSW degrades model quality in this setting (see Table 2). Elastic scaling comparisons consider one random trial; future versions of this work will include five trials.

For $S > 1$, AdaScale's learning rate increases gradually during initial training, despite the fact that `lr` is non-increasing. Unlike warm-up heuristics (Goyal et al., 2017), this behavior emerges naturally from a principled algorithm, not hand-tuned user input. Thus, AdaScale provides not only a compelling alternative to warm-up but also a plausible explanation for warm-up's success.

For `imagenet`, we also consider elastic scaling. Here, the only change to AdaScale is that $S$ changes abruptly after some iterations. We consider two cases: (i) $S$ increases from 32 to 64 at $\tau_t = T_{\mathrm{SI}}/4$ and from 64 to 128 at $\tau_t = T_{\mathrm{SI}}/2$, and (ii) the scale decreases at the same points, from 128 to 64 to 32. In Figure 4, we include training curves from this setting. AdaScale remains approximately scale invariant, highlighting AdaScale's value for the common scenario of dynamic resource availability.

**Table 2: Comparison of final model quality.** *Shorthand:* AS=AdaScale, LSW=Linear scaling rule with warm-up, gray=model quality significantly worse than for $S = 1$ (5 trials, 0.95 significance), N/A=training diverges, Elastic↑/↓=elastic scaling with increasing/decreasing scale (see Figure 4). Linear scaling leads to poor model quality as the scale increases, while AdaScale preserves model performance for nearly all cases.

| Task | $S$ | Total batch size | Validation metric | | Training loss | | Total iterations | |
|---|---|---|---|---|---|---|---|---|
| | | | AS | LSW | AS | LSW | AS | LSW |
| cifar10 | 1 | 128 | 94.1 | 94.1 | 0.157 | 0.157 | 39.1k | 39.1k |
| | 8 | 1.02k | 94.1 | 94.0 | 0.153 | 0.161 | 5.85k | 4.88k |
| | 16 | 2.05k | 94.1 | 93.6 | 0.150 | 0.163 | 3.36k | 2.44k |
| | 32 | 4.10k | 94.1 | 92.8 | 0.145 | 0.177 | 2.08k | 1.22k |
| | 64 | 8.19k | 93.9 | 76.6 | 0.140 | 0.272 | 1.41k | 611 |
| imagenet | 1 | 256 | 76.4 | 76.4 | 1.30 | 1.30 | 451k | 451k |
| | 16 | 4.10k | 76.5 | 76.3 | 1.26 | 1.31 | 33.2k | 28.2k |
| | 32 | 8.19k | 76.6 | 76.1 | 1.23 | 1.33 | 18.7k | 14.1k |
| | 64 | 16.4k | 76.5 | 75.6 | 1.19 | 1.35 | 11.2k | 7.04k |
| | 128 | 32.8k | 76.5 | 73.3 | 1.14 | 1.51 | 7.29k | 3.52k |
| | Elastic↑ | various | 76.8 | 75.7 | 1.15 | 1.36 | 11.6k | 7.04k |
| | Elastic↓ | various | 76.6 | 73.8 | 1.23 | 1.46 | 13.7k | 9.68k |
| speech | 1 | 32 | 79.6 | 79.6 | 2.03 | 2.03 | 84.8k | 84.8k |
| | 4 | 128 | 81.0 | 80.9 | 5.21 | 4.66 | 22.5k | 21.2k |
| | 8 | 256 | 80.7 | 80.2 | 6.74 | 6.81 | 12.1k | 10.6k |
| | 16 | 512 | 80.6 | N/A | 7.33 | N/A | 6.95k | 5.30k |
| | 32 | 1.02k | 80.3 | N/A | 8.43 | N/A | 4.29k | 2.65k |
| transformer | 1 | 2.05k | 27.2 | 27.2 | 1.60 | 1.60 | 1.55M | 1.55M |
| | 16 | 32.8k | 27.4 | 27.3 | 1.60 | 1.60 | 108k | 99.0k |
| | 32 | 65.5k | 27.3 | 27.0 | 1.59 | 1.61 | 58.9k | 49.5k |
| | 64 | 131k | 27.6 | 26.7 | 1.59 | 1.63 | 33.9k | 24.8k |
| | 128 | 262k | 27.4 | N/A | 1.59 | N/A | 21.4k | 12.1k |
| yolo | 1 | 16 | 80.2 | 80.2 | 2.65 | 2.65 | 207k | 207k |
| | 16 | 256 | 81.5 | 81.4 | 2.63 | 2.66 | 15.9k | 12.9k |
| | 32 | 512 | 81.3 | 80.5 | 2.61 | 2.81 | 9.27k | 6.47k |
| | 64 | 1.02k | 81.3 | 70.1 | 2.60 | 4.02 | 5.75k | 3.23k |
| | 128 | 2.05k | 81.4 | N/A | 2.57 | N/A | 4.07k | 1.62k |

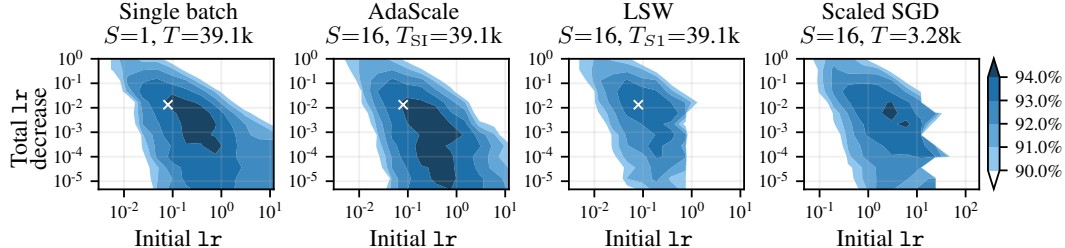

**Figure 5: Scale invariance for many learning rate schedules.** Heat maps cover the space of exponential decay `lr` schedules for `cifar10`. At scale 16, validation accuracies for AdaScale align closely with results for single-batch training, with the space of 94+% schedules growing moderately with AdaScale. With LSW, no schedule achieves 94% accuracy. On the right, direct `lr` search at scale 16 produces inferior results to AdaScale (here the total iterations, 3.28k, is the average total iterations among 94+% AdaScale trials). Thus, AdaScale induces a superior family of schedules for scaled training. The white '×' indicates the `lr` used for Figure 1.

As a baseline for all benchmarks, we also evaluate linear scaling with warm-up (LSW). As inputs, LSW takes single-batch schedule $lr_{S1} = lr$ and single-batch steps $T_{S1} = T_{SI}$, where $lr$ and $T_{SI}$ are the inputs to AdaScale. Our warm-up implementation closely follows Goyal et al. (2017). LSW trains for $\lceil T_{S1}/S \rceil$ iterations, applying warm-up to the first 5.5% of iterations. During warm-up, the learning rate increases linearly from $lr_{S1}(0)$ to $S \cdot lr_{S1}(0)$.

Table 2 compares results for AdaScale and LSW. LSW consistently trains for fewer steps, but doing so comes at a cost. As $S$ grows larger, LSW consistently degrades model quality and sometimes diverges. For these divergent cases, we also tested doubling the warm-up duration to 11% of iterations, and training still diverged. In contrast, AdaScale preserves model quality for nearly all cases.

As a final comparison, Figure 5 demonstrates AdaScale's performance on `cifar10` with many different `lr` schedules. We consider a 13×13 grid of exponential decay schedules and plot contours of resulting validation accuracy. At scale 16, AdaScale results align with accuracies for single-batch training, illustrating that AdaScale is approximately scale-invariant for many schedules. Moreover, AdaScale convincingly outperforms direct search over exponential decay schedules for scaled SGD at $S=16$. For training at scale, AdaScale provides a more natural learning rate parameterization.

## 5 SCALE-INVARIANT CONVERGENCE BOUND

We now present convergence bounds that formalize the approximate scale invariance of AdaScale. The bounds provide identical convergence guarantees for all scales, meaning that in terms of upper bounds on training loss, AdaScale is scale invariant. For comparison, we include an analogous bound for the linear scaling rule. Qualitatively, the bounds agree closely with our empirical results.

Let us define $F^* = \min_{\mathbf{w}} F(\mathbf{w})$. Our analysis requires a few assumptions that are typical of SGD analysis of non-convex problems (see, for example, (Lei et al., 2017; Yuan et al., 2019)):

**Assumption 1** ($\alpha$-Polyak-Łojasiewicz). *For some $\alpha > 0$, $F(\mathbf{w}) - F^* \leq \frac{1}{2\alpha} \|\nabla F(\mathbf{w})\|^2$ for all $\mathbf{w}$.*

**Assumption 2** ($\beta$-smooth). *For some $\beta > 0$, $\|\nabla F(\mathbf{w}) - \nabla F(\mathbf{w}')\| \leq \beta \|\mathbf{w} - \mathbf{w}'\|$ for all $\mathbf{w}$, $\mathbf{w}'$.*

**Assumption 3** (Bounded variance). *There exists a $V \geq 0$ such that $\sigma^2(\mathbf{w}) \leq V$ for all $\mathbf{w}$.*

We emphasize that we do not assume convexity. The PL condition, which is perhaps our strongest assumption, is proven to hold for some nonlinear neural networks (Charles & Papailiopoulos, 2018).

We consider constant `lr` schedules, which result in simple and instructive bounds. To provide context for the AdaScale result, we first present a straightforward bound for single-batch training:

**Theorem 1** (Single-batch SGD bound). *Given Assumptions 1, 2, 3 and $\eta \in (0, 2\beta^{-1})$, consider Algorithm 1 with $S = 1$ and $lr(t) = \eta$. Defining $\gamma = \eta\alpha(2 - \eta\beta)$ and $\Delta = \frac{1}{2\gamma}\eta^2\beta V$, we have*

$$\mathbb{E}\left[F(\mathbf{w}_T) - F^*\right] \leq (1 - \gamma)^T \left[F(\mathbf{w}_0) - F^*\right] + \Delta.$$

The bound describes two important characteristics of the single-batch algorithm. First, the suboptimality converges in expectation to at most $\Delta$. Second, convergence to $\Delta + \epsilon$ requires at most

$\lceil \log((F(\mathbf{w}_0) - F^*)\epsilon^{-1}) / \log((1-\gamma)^{-1}) \rceil$ iterations. We note similar bounds exist, under a stronger variance assumption (Karimi et al., 2016; Reddi et al., 2016; De et al., 2017; Yin et al., 2018).

Importantly, our AdaScale bound converges to this same $\Delta$ for all practical values of $S$:

**Theorem 2** (AdaScale bound). *Define $\gamma$, $\Delta$ as in Theorem 1. Given Assumptions 1, 2, 3, $S \le \gamma^{-1}$, and $\eta \in (0, 2\beta^{-1})$, define $\mathbf{w}_T$ as the result of Algorithm 2 with $\mathtt{lr}(t) = \eta$ and scale $S$. Then*

$$\mathbb{E}\left[F(\mathbf{w}_T) - F^*\right] \le (1-\gamma)^{T_{\mathrm{SI}}} \left[F(\mathbf{w}_0) - F^*\right] + \Delta.$$

This bound for AdaScale is scale invariant, as it does not depend on $S$. Like single-batch SGD, the suboptimality converges in expectation to at most $\Delta$, but AdaScale achieves this for all scales. In addition, AdaScale speeds up training by a factor $\bar{r} = \frac{1}{T} \sum_{t=0}^{T-1} r_t$. That is, convergence to $\Delta + \epsilon$ requires at most $\lceil \bar{r}^{-1} \log((F(\mathbf{w}_0) - F^*)\epsilon^{-1}) / \log((1-\gamma)^{-1}) \rceil$ iterations (since $T_{\mathrm{SI}} \le \tau_T = \bar{r}T$).

As a final comparison, we provide an analogous bound for linear scaling, which is not scale invariant:

**Theorem 3** (Bound for linear scaling rule). *Define $\gamma$ and $\Delta$ as in Theorem 1. Given Assumptions 1, 2, 3, $S \le \gamma^{-1}$, and $\eta \in (0, 2(S\beta)^{-1})$, consider Algorithm 1 with $\mathtt{lr}(t) = S\eta$. We have*

$$\mathbb{E}\left[F(\mathbf{w}_T) - F^*\right] \le \left(1 - \gamma \cdot \left(\frac{2 - S\eta\beta}{2 - \eta\beta}\right)\right)^{ST} \left[F(\mathbf{w}_0) - F^*\right] + \left(\frac{2 - \eta\beta}{2 - S\eta\beta}\right) \Delta.$$

Unlike Theorem 2, this bound converges to a value that increases with $S$. In addition, a smaller range of learning rates guarantees convergence. In practical terms, this means that linear scaling often leads to worse model quality and greater risk of divergence, especially for large $S$. These differences appear throughout our empirical comparisons in §4.

Finally, we note both Theorem 2 and Theorem 3 require that $S \le \gamma^{-1}$. For practical problems, $\gamma$ is small, and we can safely ignore this constraint. Otherwise single batch training would converge quickly, due to Theorem 1, and smaller scales would result in fast training.

## 6 RELATION TO PRIOR WORK

While linear scaling with warm-up is perhaps the most popular scaling rule, researchers have considered a few alternative strategies. "Square root learning rate scaling" (Krizhevsky, 2014; Li et al., 2014; Hoffer et al., 2017; You et al., 2018) multiplies learning rates by the square root of the batch size increase. Across scales, this preserves the covariance of the SGD update. Establishing this invariant remains poorly justified, however, and often root scaling degrades model quality in practice (Goyal et al., 2017; Golmant et al., 2018; Jastrzębski et al., 2018). AdaScale adapts learning rates by making $\eta_t \mathbb{E}\left[\|\bar{\mathbf{g}}_t\|^2\right]$ invariant across scales, which results in our scale-invariant bound from §5. Finally, we might also consider model-specific scaling rules, such as LARS for CNNs (You et al., 2017). AdaScale solves the general problem (P1), making AdaScale applicable to many models.

Many prior works have also considered the role of gradient variance in SGD. McCandlish et al. (2018) study the impact of gradient variance on scaling efficiency. These general findings also apply to AdaScale, as gradient variance similarly determines AdaScale's efficiency. Much like AdaScale, Johnson & Guestrin (2018) also adapt learning rates to lower amounts of gradient variance—in this case when using SGD with importance sampling. Because the variance reduction is relatively small in this setting, however, distributed training can have far greater impact on training times. Lastly, many algorithms also adapt to gradient moments for improved training, given a fixed amount of variance—see (Schaul et al., 2013; Kingma & Ba, 2015; Balles & Hennig, 2018), just to name a few. AdaScale adapts learning rates across scales, which correspond to different amounts of gradient variance. Perhaps future algorithms will combine approaches in order to achieve both goals.

## 7 DISCUSSION

SGD is not perfectly parallelizable. Unsurprisingly, the linear scaling rule can fail at large scales. In contrast, AdaScale accepts sublinear speedups in order to better preserve model quality. What do the speed-ups from AdaScale tell us about the scaling efficiency of SGD in general? For many problems, such as `imagenet` with batch size 32.8k, AdaScale establishes lower bounds on SGD's scaling efficiency. An important remaining question is whether AdaScale is optimally efficient, or if other practical algorithms can achieve similar scale invariance with fewer iterations.

AdaScale provides a useful new parameterization of learning rate schedules for large-batch SGD. We provide a simple `lr` schedule, which AdaScale adapts to learning rates for scaled training. From this, warm-up behavior emerges naturally, which produces quality models for many problems and scales. Even in elastic scaling settings, AdaScale adapts successfully to the state of training. Given these appealing qualities, it seems important to further study such learning rate schedules.

Based on our empirical results, as well as the algorithm's practicality and theoretical justification, we believe that AdaScale is valuable for speeding up training in practice.

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

# A    PROOFS

In this appendix, we prove the results from §5 and §3. We first prove a lemma in §A.1, which we apply in the proofs. We prove Theorem 1 in §A.2, Theorem 2 in §A.3, and Theorem 3 in §A.4. We also prove Proposition 1 in §A.5 and Proposition 2 in §A.6.

## A.1    KEY LEMMA

**Lemma 1.** *Given Assumptions 1, 2, 3 and $\eta \in (0, 2\beta^{-1})$, define $\gamma = \eta\alpha(2-\eta\beta)$ and $\Delta = \frac{1}{2\gamma}\eta^2\beta V$. Consider Algorithm 2 with $\mathrm{lr}(t) = \eta$. For all iterations $t$, we have*

$$\mathbb{E}\left[F(\mathbf{w}_t) - F^*\right] \leq [F(\mathbf{w}_0) - F^*]\prod_{t'=0}^{t-1}(1 - r_{t'}\gamma) + \Delta\,.$$

*Proof.* We prove this by induction. To simplify notation, let us define $\tilde{F}(\mathbf{w}) = F(\mathbf{w}) - F^*$. For $t = 0$, we have

$$\mathbb{E}[\tilde{F}(\mathbf{w}_0)] = \tilde{F}(\mathbf{w}_0) \leq \tilde{F}(\mathbf{w}_0)\prod_{t'=0}^{-1}(1 - r_{t'}\gamma) + \Delta\,.$$

Note we are using the convention $\prod_{i=0}^{-1}x_i = 1$. For $t \geq 1$, assume the inductive hypothesis

$$\mathbb{E}[\tilde{F}(\mathbf{w}_{t-1})] \leq \tilde{F}(\mathbf{w}_0)\prod_{t'=0}^{t-2}(1 - r_{t'}\gamma) + \Delta\,. \tag{1}$$

Applying Assumption 2 (smoothness) and the update equation $\mathbf{w}_t = \mathbf{w}_{t-1} - r_{t-1}\eta\bar{\mathbf{g}}_{t-1}$, we have

$$\begin{aligned}\tilde{F}(\mathbf{w}_t) &\leq \tilde{F}(\mathbf{w}_{t-1}) + \langle\nabla F(\mathbf{w}_{t-1}), \mathbf{w}_t - \mathbf{w}_{t-1}\rangle + \frac{\beta}{2}\|\mathbf{w}_t - \mathbf{w}_{t-1}\|^2 \\ &= \tilde{F}(\mathbf{w}_{t-1}) - r_{t-1}\eta\langle\nabla F(\mathbf{w}_{t-1}), \bar{\mathbf{g}}_{t-1}\rangle + r_{t-1}^2\eta^2\frac{\beta}{2}\|\bar{\mathbf{g}}_{t-1}\|^2\,.\end{aligned}$$

Taking the expectation with respect to the $S$ random batches from step $t$, we have

$$\mathbb{E}\left[\tilde{F}(\mathbf{w}_t) \mid \mathbf{w}_{t-1}\right] \leq \tilde{F}(\mathbf{w}_{t-1}) - r_{t-1}\eta\|\nabla F(\mathbf{w}_{t-1})\|^2 + r_{t-1}^2\eta^2\frac{\beta}{2}\mathbb{E}\left[\|\bar{\mathbf{g}}_{t-1}\|^2 \mid \mathbf{w}_{t-1}\right]\,.$$

Now taking the expectation with respect to the distribution of $\mathbf{w}_{t-1}$, it follows that

$$\mathbb{E}\left[\tilde{F}(\mathbf{w}_t)\right] \leq \mathbb{E}\left[\tilde{F}(\mathbf{w}_{t-1})\right] - r_{t-1}\eta\mathbb{E}\left[\|\nabla F(\mathbf{w}_{t-1})\|^2\right] + r_{t-1}^2\eta^2\frac{\beta}{2}\mathbb{E}\left[\|\bar{\mathbf{g}}_{t-1}\|^2\right]\,. \tag{2}$$

For the last term, we have

$$\begin{aligned}\mathbb{E}\left[\|\bar{\mathbf{g}}_{t-1}\|^2\right] &= \mathbb{E}\left[\|(\bar{\mathbf{g}}_{t-1} - \nabla F(\mathbf{w}_{t-1})) + \nabla F(\mathbf{w}_{t-1})\|^2\right] \\ &= \mathbb{E}\left[\|\bar{\mathbf{g}}_{t-1} - \nabla F(\mathbf{w}_{t-1})\|^2 + \|\nabla F(\mathbf{w}_{t-1})\|^2\right] \\ &= \mathbb{E}\left[\frac{1}{S}\sigma^2(\mathbf{w}_{t-1}) + \|\nabla F(\mathbf{w}_{t-1})\|^2\right] \\ &= \frac{1}{r_{t-1}}\mathbb{E}\left[\sigma^2(\mathbf{w}_{t-1}) + \|\nabla F(\mathbf{w}_{t-1})\|^2\right] \\ &\leq \frac{1}{r_{t-1}}\left(\mathbb{E}\left[\|\nabla F(\mathbf{w}_{t-1})\|^2\right] + V\right)\,.\end{aligned} \tag{3}$$

Combining (3) with (2), we have

$$\begin{aligned}\mathbb{E}\left[\tilde{F}(\mathbf{w}_t)\right] &\leq \mathbb{E}\left[\tilde{F}(\mathbf{w}_{t-1})\right] - r_{t-1}\eta(1 - \eta\frac{\beta}{2})\mathbb{E}\left[\|\nabla F(\mathbf{w}_{t-1})\|^2\right] + r_{t-1}\eta^2\frac{\beta}{2}V \\ &\leq (1 - r_{t-1}\gamma)\mathbb{E}\left[\tilde{F}(\mathbf{w}_{t-1})\right] + r_{t-1}\gamma\Delta\,.\end{aligned} \tag{4}$$

In this last step, we applied Assumption 1 (PL condition) and plugged in definitions for $\gamma$ and $\Delta$.

To complete the proof, we apply (1):

$$\mathbb{E}[\tilde{F}(\mathbf{w}_t)] \le (1 - r_{t-1}\gamma)\left(\tilde{F}(\mathbf{w}_0)\prod_{t'=0}^{t-2}(1 - r_{t'}\gamma) + \Delta\right) + r_{t-1}\gamma\Delta$$

$$= \tilde{F}(\mathbf{w}_0)\prod_{t'=0}^{t-1}(1 - r_{t'}\gamma) + \Delta.$$

$\square$

## A.2 Proof of Theorem 1

**Theorem 1** (Single-batch SGD bound). *Given Assumptions 1, 2, 3 and $\eta \in (0, 2\beta^{-1})$, consider Algorithm 1 with $S = 1$ and $\mathrm{lr}(t) = \eta$. Defining $\gamma = \eta\alpha(2 - \eta\beta)$ and $\Delta = \frac{1}{2\gamma}\eta^2\beta V$, we have*

$$\mathbb{E}\left[F(\mathbf{w}_T) - F^*\right] \le (1 - \gamma)^T\left[F(\mathbf{w}_0) - F^*\right] + \Delta.$$

*Proof.* The theorem is a special case of Lemma 1. In particular, Algorithm 1 with inputs $\mathrm{lr}(t) = \eta$, $S = 1$, and $T$ iterations is equivalent to Algorithm 2 with $T_{\mathrm{SI}} = T$ and the same scale and learning rate inputs. This follows from the fact that $r_t = 1$ for all iterations of AdaScale when $S = 1$. Thus, we can obtain the result by plugging $t = T$ into the bound from Lemma 1. $\square$

## A.3 Proof of Theorem 2

**Theorem 2** (AdaScale bound). *Define $\gamma$, $\Delta$ as in Theorem 1. Given Assumptions 1, 2, 3, $S \le \gamma^{-1}$, and $\eta \in (0, 2\beta^{-1})$, define $\mathbf{w}_T$ as the result of Algorithm 2 with $\mathrm{lr}(t) = \eta$ and scale $S$. Then*

$$\mathbb{E}\left[F(\mathbf{w}_T) - F^*\right] \le (1 - \gamma)^{T_{\mathrm{SI}}}\left[F(\mathbf{w}_0) - F^*\right] + \Delta.$$

*Proof.* Let $T$ denote the total iterations for Algorithm 2. Applying Lemma 1, we have

$$\mathbb{E}\left[F(\mathbf{w}_t) - F^*\right] \le (F(\mathbf{w}_0) - F^*)\prod_{t'=0}^{T-1}(1 - r_{t'}\gamma) + \Delta. \tag{5}$$

Now note that for any $r \ge 1$ and $x \in [0, 1]$, we have

$$1 - rx \le (1 - x)^r. \tag{6}$$

This holds because for any $r \ge 1$ and $x \in [0, 1]$, the function $(1 - x)^r$ is convex in $x$, and $1 - rx$ is tangent to this function at $x = 0$. Thus,

$$\prod_{t'=0}^{T-1}(1 - r_{t'}\gamma) \le (1 - \gamma)^{\sum_{t'=0}^{T-1} r_{t'}}. \tag{7}$$

Note that this requires $1 - r_t\gamma \ge 0$ for all $t$, which is true because $r_t \le S \le \gamma^{-1}$. Now plugging (7) into (5),

$$\mathbb{E}\left[F(\mathbf{w}_t) - F^*\right] \le (F(\mathbf{w}_0) - F^*)(1 - \gamma)^{\sum_{t'=0}^{T-1} r_{t'}} + \Delta$$

$$\le (F(\mathbf{w}_0) - F^*)(1 - \gamma)^{T_{\mathrm{SI}}} + \Delta.$$

In the last step, we use the stopping condition of Algorithm 2 ($T_{\mathrm{SI}} \le \tau_T = \sum_{t=0}^{T-1} r_t$). $\square$

## A.4 Proof of Theorem 3

**Theorem 3** (Bound for linear scaling rule). *Define $\gamma$ and $\Delta$ as in Theorem 1. Given Assumptions 1, 2, 3, $S \le \gamma^{-1}$, and $\eta \in (0, 2(S\beta)^{-1})$, consider Algorithm 1 with $\mathrm{lr}(t) = S\eta$. We have*

$$\mathbb{E}\left[F(\mathbf{w}_T) - F^*\right] \le \left(1 - \gamma \cdot \left(\frac{2 - S\eta\beta}{2 - \eta\beta}\right)\right)^{ST}\left[F(\mathbf{w}_0) - F^*\right] + \left(\frac{2 - \eta\beta}{2 - S\eta\beta}\right)\Delta.$$

*Proof.* We reduce the theorem to a special case of Theorem 1. Define $\tilde{\mathbf{x}} = (\tilde{\mathbf{x}}^{(1)}, \ldots, \tilde{\mathbf{x}}^{(S)})$, where $\tilde{\mathbf{x}}^{(i)} \sim \mathcal{X}$ for each $i \in [S]$, and $\tilde{\mathbf{x}}^{(1)}, \ldots, \tilde{\mathbf{x}}^{(S)}$ are jointly independent. Denote by $\tilde{\mathcal{X}}$ the distribution of $\tilde{\mathbf{x}}$. Also define

$$\tilde{f}(\mathbf{w}, \tilde{\mathbf{x}}) = \frac{1}{S} \sum_{i=1}^{S} f(\mathbf{w}, \tilde{\mathbf{x}}^{(i)}).$$

It follows that for any $\mathbf{w}$,

$$\mathbb{E}_{\tilde{\mathbf{x}}} \left[ \|\nabla \tilde{f}(\mathbf{w}, \tilde{\mathbf{x}}) - \nabla F(\mathbf{w})\|^2 \right] = \frac{1}{S} \sigma^2(\mathbf{w}) \leq \frac{V}{S}.$$

The algorithm described in Theorem 3 is identical to running Algorithm 1 with scale 1, batch distribution $\tilde{\mathcal{X}}$, loss $\tilde{f}$, learning rate $\mathtt{lr}(t) = S\eta$, and variance upper bound $\frac{V}{S}$. Plugging these values into Theorem 1, we have

$$\mathbb{E}\left[F(\mathbf{w}_T) - F^*\right] \leq (1 - S\eta\alpha(2 - S\eta\beta))^T [F(\mathbf{w}_0) - F^*] + \frac{S\eta\beta V S^{-1}}{2\alpha(2 - S\eta\beta)}$$

$$= \left(1 - S\gamma \cdot \left(\frac{2 - S\eta\beta}{2 - \eta\beta}\right)\right)^T [F(\mathbf{w}_0) - F^*] + \left(\frac{2 - \eta\beta}{2 - S\eta\beta}\right)\Delta$$

$$\leq \left(1 - \gamma \cdot \left(\frac{2 - S\eta\beta}{2 - \eta\beta}\right)\right)^{ST} [F(\mathbf{w}_0) - F^*] + \left(\frac{2 - \eta\beta}{2 - S\eta\beta}\right)\Delta.$$

The last step follows from (6). $\qquad \square$

### A.5 PROOF OF PROPOSITION 1

**Proposition 1** (Scale-invariant SGD for deterministic gradients). *If $\sigma^2(\mathbf{w}) = 0$ for all $\mathbf{w} \in \mathbb{R}^d$, then applying identity scaling to Algorithm 1 results in a scale-invariant algorithm.*

*Proof.* Since the gradient variance is zero, the `compute_gradient` function returns $\nabla F(\mathbf{w}_t)$, which does not depend on $S$. Thus, the algorithm does not depend on $S$ in this case, which implies that it is scale-invariant. $\qquad \square$

### A.6 PROOF OF PROPOSITION 2

**Proposition 2** (Scale-invariant SGD for extreme stochasticity). *Consider fixed covariance matrix $\tilde{\mathbf{\Sigma}} \in \mathbb{S}_{++}^d$, learning rate value $\tilde{\eta} \in \mathbb{R}_{>0}$, and training duration $\tilde{T}$. For a given $\nu \in \mathbb{R}_{>0}$, assume $\nabla_{\mathbf{w}} f(\mathbf{w}, \mathbf{x}) \sim \mathcal{N}(\nabla F(\mathbf{w}), \nu\tilde{\mathbf{\Sigma}})$, and apply linear scaling to Algorithm 1 with $\mathtt{lr}_{S1}(t) = \nu^{-1}\tilde{\eta}$ and $T_{S1} = \nu\tilde{T}$. The resulting scaled SGD algorithm is scale-invariant in the limit $\nu \to +\infty$.*

*Proof.* The scaled SGD algorithm runs for $\lceil \nu\tilde{T}/S \rceil$ iterations and follows the update rule

$$\mathbf{w}_{t+1} = \mathbf{w}_t - \frac{S\tilde{\eta}}{\nu}\nabla F(\mathbf{w}_t) + \frac{S\tilde{\eta}}{\nu}\boldsymbol{\xi}_t.$$

Here $\boldsymbol{\xi}_t$ is normally distributed with $\mathbb{E}[\boldsymbol{\xi}_t] = \mathbf{0}$ and $\mathrm{cov}(\boldsymbol{\xi}_t, \boldsymbol{\xi}_t) = \frac{\nu}{S}\tilde{\mathbf{\Sigma}}$. In the limit $\nu \to +\infty$, this difference equation converges to a stochastic differential equation on the interval $[0, \tilde{\eta}\tilde{T}]$ (Kloeden & Platen, 1992, Chapter 9):

$$d\mathbf{w} = -\nabla F(\mathbf{w})dt + (\tilde{\eta}\tilde{\mathbf{\Sigma}})^{1/2}d\mathbf{W}(t), \quad \text{where} \quad \mathbf{W}(t) \sim \mathcal{N}(\mathbf{0}, \mathbf{I}).$$

Since this SDE does not depend on $S$, the algorithm is scale-invariant in this limit. $\qquad \square$

## B ADDITIONAL DETAILS ON EMPIRICAL COMPARISONS

This appendix provides additional details of our experiment set-up.

## B.1 LEARNING RATE SCHEDULES

We describe the `lr` schedules for each training benchmark in Table 3. We use two learning rate families: exponential decay and step decay. Using parameters $\eta_0$, $d$, and $w_i$, we define $\texttt{lr}(t) = \eta_0 d^{(t/T_{S1})}$ for exponential decay families and $\texttt{lr}(t) = \eta_0 d^{\sum_i^n \mathbb{1}[t>w_i]}$ for step decay families. Here $T_{S1}$ denotes the total iterations for scale $S = 1$. Note that in all cases, we use simple schedules and no warm-up.

**Table 3: Learning rate schedules for training benchmarks.**

| Benchmark | Learning rate famliy | $\eta_0$ | $d$ | $w_i$ |
|---|---|---|---|---|
| `cifar10` | Exponential decay | 0.08 | 0.0133 | N/A |
| `imagenet` | Step decay | 0.1 | 0.1 | 150,240, 300,480, 400,640 |
| `speech` | Exponential decay | $1.4 \times 10^{-3}$ | 0.05 | N/A |
| `transformer` | Step decay | 0.01 | 0.1 | 1,440,000 |
| `yolo` | Step decay | $2.5 \times 10^{-4}$ | 0.1 | 160,000, 180,000 |

For `imagenet` and `yolo`, we used standard learning rate schedules from (Goyal et al., 2017) and (Zhang et al., 2019). For `cifar10`, `speech`, and `transformer`, we chose learning rate parameters, via hand-tuning, that approximately maximized model quality. This was necessary for `speech` and `transformer`, since our reference implementations train with the Adam optimizer (Kingma & Ba, 2015), and momentum-SGD requires different learning rate values.

## B.2 WARM-UP IMPLEMENTATION

Our warm-up procedure closely follows the strategy of Goyal et al. (2017). We apply warm-up for the first 5.5% of training iterations—we denote this number by $W_S$. During warm-up, the learning rate increases linearly, starting at the initial learning rate for single-batch training and finishing at $S$ times this value. After warm-up, we apply linear scaling to the single-batch schedule. Following Goyal et al. (2017), we modify this scaled schedule so that the total iterations, including warm-up, is proportional to $S^{-1}$. For step-decay schedules, we omit the first $W_S$ iterations after warm-up. For exponential decay schedules, we compress the scaled schedule by $W_S$ iterations, using slightly faster decay.

## B.3 BENCHMARK-SPECIFIC IMPLEMENTATION DETAILS

Here we describe implementation details that are specific to each benchmark task.

### B.3.1 `cifar10`

We train ResNet-18 (preactivation) models (He et al., 2016b), using the standard training data split for CIFAR-10 (Krizhevsky, 2009). We use weight decay $= 5 \times 10^{-4}$. For batch normalization, we use parameters momentum $= 0.995$ and $\epsilon = 2 \times 10^{-5}$, and we do not train the batch normalization scaling parameters. We apply standard data augmentation during training. Specifically, we pad images to $40 \times 40$ and random crop to $32 \times 32$, and we also apply random horizontal reflections.

### B.3.2 `imagenet`

For ImageNet classification (Deng et al., 2009), we train ResNet-50 models (He et al., 2016a). Our implementation closely follows the implementation of Goyal et al. (2017). We use stride-2 convolutions on $3 \times 3$ layers. For each block's final batch normalization layer, we initialize the batch norm scaling parameters to $0$ (and we initialize to $1$ everywhere else). We use weight decay parameter $10^{-4}$. Since each GPU processes 128 examples per batch, we use ghost batch normalization (Hoffer et al., 2017) with ghost batch size 32. We resize input images to $224 \times 224 \times 3$. For data augmentation, we apply random cropping and left-right mirroring during training.

### B.3.3 `speech`

We use Amodei et al. (2016)'s Deep Speech 2 model architecture. The model consists of two 2D convolutional input layers, five bidirectional RNN layers, one fully connected layer, and softmax outputs. Each convolutional layer has 32 filters. The RNN layers use GRU cells with hidden size 800. We apply batch normalization to the inputs of each layer. The batch norm parameters are momentum $= 0.997$ and $\epsilon = 10^{-5}$. The loss is CTC loss. The inputs to the network are log spectrograms, which we compute using 20ms windows from audio waveforms sampled at 16 kHz. The training data is the `train-clean-100` and `train-clean-360` partitions of the OpenSLR LibriSpeech Corpus, which amounts to 460 hours of recorded speech. We evaluate models on the `dev-clean` partition.

### B.3.4 `transformer`

We train Transformer base models (Vaswani et al., 2017). We use dynamic batching with at most 256 tokens per example. In Table 2, the "batch size" is the maximum number of tokens processed per iteration. Our implementation closely follows that of Vaswani et al. (2017). Unlike Vaswani et al., we use only the final model for evaluation instead of the average of the last five checkpoints. We train on the WMT 2014 English-German dataset and evaluate on the `newstest2014` test set.

### B.3.5 `yolo`

We train YOLOv3 models (Redmon & Farhadi, 2018). To achieve high mAP scores, we also apply mixup (Zhang et al., 2018) and class label smoothing, following (Zhang et al., 2019). We also use focal loss. We use batch normalization momentum$= 0.9$ and weight decay $= 5 \times 10^{-4}$. We resize input images to $416 \times 416$ (for both training and validation). We report mAP values at IOU threshold $0.5$. We use the Pascal VOC 2007 `trainval` and 2012 `trainval` datasets for training and the 2007 test set for validation (Everingham et al., 2010). During training, we initialize the darknet-53 convolutional layers with weights trained on ImageNet.

### B.4 MISCELLANEOUS

In practice, wall time speed-ups also depend on system scaling efficiency. Since most aspects of system scaling relate orthogonally to the training algorithm, we limit our scope to algorithmic aspects of training.

For Figure 5, one dimension defines initial value `lr(0)`, and the second dimension specifies total decrease $\texttt{lr}(T_{\mathrm{SI}})/\texttt{lr}(0)$. For single-batch training, we use $T = 39.1 \times 10^3$ steps. We run AdaScale and the LW baseline at $S = 16$, and we compare the final validation accuracies.

## C ROBUSTNESS TO AVERAGING PARAMETER

In this appendix, we test the robustness of AdaScale to the averaging parameter $\theta$ for estimating gain ratios (see §3.3). When $\theta = 0$, AdaScale does not average estimates of gradient moments. The closer $\theta$ is to 1, the more that AdaScale averages across iterations.

Using the `cifar10` benchmark, we compare four values of $\theta$ at scales $S = 8$ and $S = 32$. The case $\theta = 1 - S/1000$ corresponds to the `cifar10` experiment for Figure 1. We average the resulting metrics over five trials. Figure 6 contains the training curves.

We also include final metric values in Table 4.

For the three smaller settings of $\theta$, the results align very closely. This suggests that AdaScale is robust to the choice of $\theta$. When $\theta = 1 - S/10000$, we see that smoothing more significantly biases gain ratio estimates, which leads to more contrasting results.

## D ADDITIONAL EMPIRICAL RESULTS

This appendix provides additional empirical results.

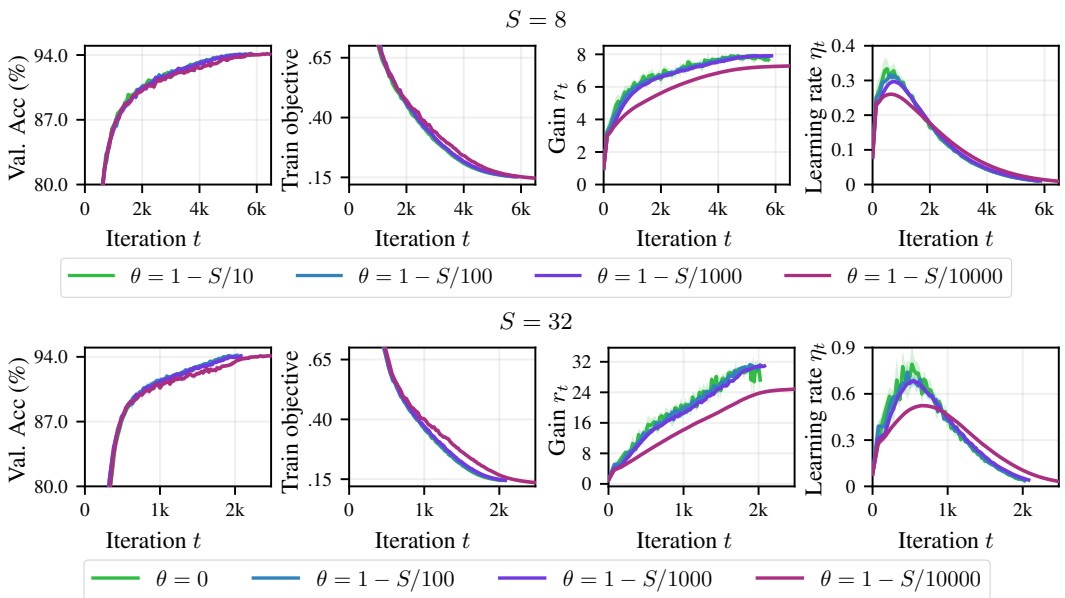

Figure 6: AdaScale training curves with varying moving average parameter.

Table 4: AdaScale final metrics with varying moving average parameter.

| $S$ | $\theta$ | Final val. accuracy (%) | Final train objective | Total iterations |
|---|---|---|---|---|
| 8 | $1 - S/10$ | 94.0 | 0.153 | 5.75k |
|  | $1 - S/100$ | 94.1 | 0.154 | 5.78k |
|  | $1 - S/1000$ | 94.1 | 0.153 | 5.85k |
|  | $1 - S/10000$ | 94.1 | 0.147 | 6.45k |
| 32 | $0$ | 94.0 | 0.145 | 2.02k |
|  | $1 - S/100$ | 94.1 | 0.147 | 2.03k |
|  | $1 - S/1000$ | 94.1 | 0.145 | 2.08k |
|  | $1 - S/10000$ | 94.1 | 0.136 | 2.46k |

### D.1 EFFECT OF NUMBER OF STEPS ON LINEAR SCALING WITH WARMUP

As can be seen from empricial results in §4, AdaScale takes more steps than LSW for all scales. In order to understand the contribution of increased step number to the improved performance of Ada-Scale, we compare AdaScale to another algorithm called LSW+. LSW+ runs for the same number of steps as AdaScale. It scales the steps axis of the LSW learning rate schedule while keeping the learning rate axis the same. Specifically, it takes single-batch schedule $\texttt{lr}_{S1} = \texttt{lr}$ and steps $T_{S1} = T_{AS}$ as inputs, where $T_{AS}$ is the average number of iterations (over five trials) taken by AdaScale. It applys warm-up to the first 5.5% of the $T_{AS}$ iterations. During warm-up, the learning rate increases linearly from $\texttt{lr}_{S1}(0)$ to $S \cdot \texttt{lr}_{S1}(0)$.

As can be seen from Table 5, the behavior of LSW+ is generally similar to that of LSW. As expected, LSW+ improves upon LSW, but LSW+ still degrades model quality at larger scales for all problems. For `speech`, `transformer`, and `yolo`, LSW+ diverges at the largest scales.

We also note that LSW+ is not a practical algorithm, because it requires either (i) first running AdaScale to determine the number of iterations; or (ii) tuning the number of iterations. The second path is inconvenient in practice. Moreover, it seems for a fair comparison, we would also need to consider AdaScale with tuning. Thus, even if LSW+ matched AdaScale, which it does not, AdaScale would still be preferable to LSW+.

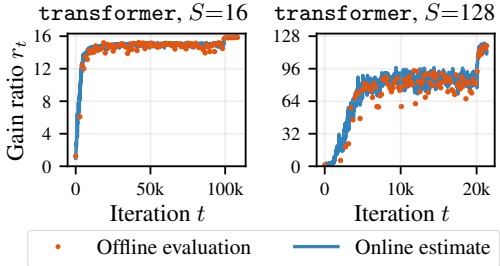

**Figure 7: Gain ratios for** `transformer`. Plots compare moving average $r_t$ estimates to values computed offline (using 1000 batches).

## D.2    GAIN RATIO ESTIMATION

Our online gain ratio estimates align closely with offline estimates (computed by averaging over 1000 batches). Figure 7 demonstrates this for the `transformer` task.

## D.3    `cifar10` SCALE INVARIANCE CURVES

Figure 8 shows additional plots for the `cifar10` task. Notably, training loss curves at various scales and full view of the learning rate curves are shown.

**Table 5: Comparison of AS and LSW+.** *Shorthand:* AS=AdaScale, LSW+=Stretched linear scaling rule with warm-up, takes the same number of steps as AS gray=model quality significantly worse than for $S = 1$ (5 trials, 0.95 significance), N/A=training diverges

| Task | $S$ | Total batch size | Validation metric | | Training loss | | Total iterations | |
|---|---|---|---|---|---|---|---|---|
| | | | AS | LSW+ | AS | LSW+ | AS | LSW+ |
| `cifar10` | 1 | 128 | 94.1 | 94.1 | 0.157 | 0.157 | 39.1k | 39.1k |
| | 8 | 1.02k | 94.1 | 94.0 | 0.153 | 0.145 | 5.85k | 5.85k |
| | 16 | 2.05k | 94.1 | 94.1 | 0.150 | 0.136 | 3.36k | 3.36k |
| | 32 | 4.10k | 94.1 | 94.0 | 0.145 | 0.128 | 2.08k | 2.08k |
| | 64 | 8.19k | 93.9 | 93.0 | 0.140 | 0.128 | 1.41k | 1.41k |
| `imagenet` | 1 | 256 | 76.4 | 76.4 | 1.30 | 1.30 | 451k | 451k |
| | 16 | 4.10k | 76.5 | 76.5 | 1.26 | 1.27 | 33.2k | 33.2k |
| | 32 | 8.19k | 76.6 | 76.4 | 1.23 | 1.24 | 18.7k | 18.7k |
| | 64 | 16.4k | 76.5 | 76.5 | 1.19 | 1.20 | 11.2k | 11.2k |
| | 128 | 32.8k | 76.5 | 75.5 | 1.14 | 1.20 | 7.29k | 7.29k |
| `speech` | 1 | 32 | 79.6 | 79.6 | 2.03 | 2.03 | 84.8k | 84.8k |
| | 4 | 128 | 81.0 | 81.0 | 5.21 | 4.22 | 22.5k | 22.5k |
| | 8 | 256 | 80.7 | 80.7 | 6.74 | 6.61 | 12.1k | 12.1k |
| | 16 | 512 | 80.6 | N/A | 7.33 | N/A | 6.95k | 6.95k |
| | 32 | 1.02k | 80.3 | N/A | 8.43 | N/A | 4.29k | 4.29k |
| `transformer` | 1 | 2.05k | 27.2 | 27.2 | 1.60 | 1.60 | 1.55M | 1.55M |
| | 16 | 32.8k | 27.4 | 27.4 | 1.60 | 1.59 | 108k | 108k |
| | 32 | 65.5k | 27.3 | 27.3 | 1.59 | 1.59 | 58.9k | 58.9k |
| | 64 | 131k | 27.6 | 27.1 | 1.59 | 1.60 | 33.9k | 33.9k |
| | 128 | 262k | 27.4 | N/A | 1.59 | N/A | 21.4k | 21.4k |
| `yolo` | 1 | 16 | 80.2 | 80.2 | 2.65 | 2.65 | 207k | 207k |
| | 16 | 256 | 81.5 | 81.9 | 2.63 | 2.47 | 15.9k | 15.9k |
| | 32 | 512 | 81.3 | 81.7 | 2.61 | 2.42 | 9.27k | 9.27k |
| | 64 | 1.02k | 81.3 | 80.6 | 2.60 | 2.51 | 5.75k | 5.75k |
| | 128 | 2.05k | 81.4 | N/A | 2.57 | N/A | 4.07k | 4.07k |

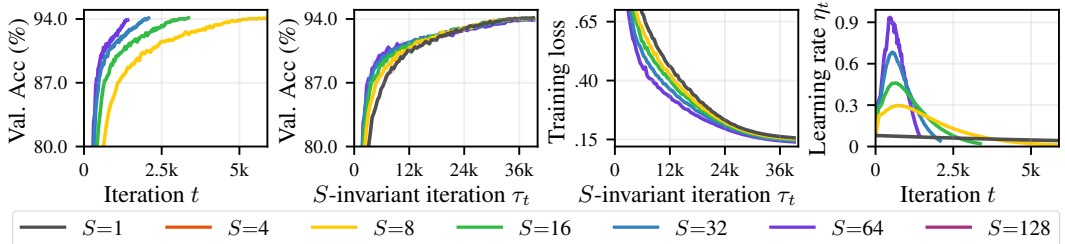

**Figure 8: AdaScale training curves for `cifar10`.** AdaScale trains quality models at various scales.

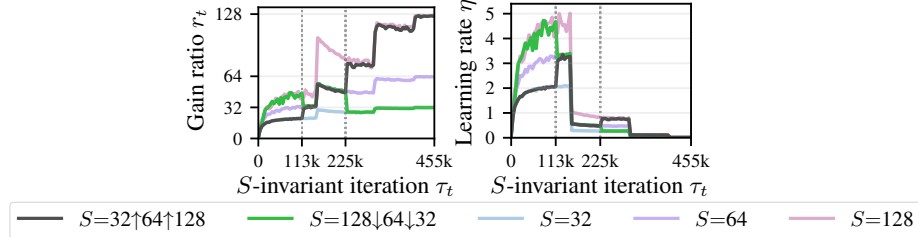

**Figure 9: Learning rate adaptation for elastic AdaScaling.** Gain ratio and learning rate curves for elastic scaling scenarios align with the corresponding curves for constant scaling scenarios, despite abrupt scale changes. (at $\tau_t = 133k$, 225k, dotted lines)

## D.4 ELASTIC SCALING

Learning rate and gain ratio curves for the two dynamic scaling scenarios we consider (discussed in §4) align surprisingly well with the corresponding curves for the scenarios where the scale is kept constant throughout the training. This is shown in Figure 9. The abrupt change in scale causes the gain ratio to change quickly which in turn leads to an almost immediate change in learning rate. This allows the algorithm to quickly adapt to varying scales.

