# OpenReview forum: "AdaScale SGD: A Scale-Invariant Algorithm for Distributed Training"
_ICLR.cc/2020/Conference — Reject_

### Official Review · AnonReviewer1 · 2019-10-15
**Official Blind Review #1**

**Rating:** 3

**Review:**

This paper proposes a novel rule for scaling the learning rate, called the gain ratio, for when the effective batch size (induced by synchronous distributed SGD) is increased. This rule for the gain ratio uses estimates of the trace of the covariance and the norm of the true gradient to determine the appropriate steplength. This results in a method with a R-linear convergence guarantee to a neighborhood that is not dependent on $S$ (which is called scale-invariant). The algorithm additionally tracks the sum of the gain ratios in order to determine the "effective" number of iterations taken, and cut the steplength appropriately.

Strengths:

The gain ratio proposed in this paper is intuitive. I particularly like how the algorithm estimates the mean and variance information in an unbiased manner to determine an appropriate scaling for the steplength. The method is able to attain a R-linear rate of convergence and appears to perform well in practice for a wide variety of applications. The gain ratio is simple to estimate within a distributed framework.

Weaknesses:

I found some of the terms in the paper to be unclear or ill-defined. The original use of the term "scale" was unclear to me. Does this refer to the number of nodes in the distributed implementation? What is its relationship to batch size?

I found the definition of scale invariance in this paper to also be unclear on first read. The claim is that the algorithm is scale invariant if its final model does not depend on $S$. What is the "final model"? As currently defined, the current analysis does not guarantee that the algorithm will reach the same final model (assuming that $f(w, x) = \ell(h(w, x), y)$, i.e. a composition of a loss function and model), as the PL condition only ensures that one reaches a global minimum, which may not be unique. In fact, the analysis only guarantees convergence to a neighborhood. The description within the analysis appears to imply that scale-invariance is a property of the algorithm attached to its convergence property. Is this the case?

The definition of scale invariance is also already used in optimization to mean algorithms that are not modified when the objective is multiplied by a constant or an affine transformation. This adds to the lack of clarity, and I would suggest the authors use a different term for this kind of invariance (batch size invariant, or something like that?).

Is the theoretical comparison between SGD and AdaScale fair? Note that one can prove a stronger convergence result with SGD because one can actually attain a Q-linear rate of convergence to a neighborhood (for a proof, see for example, Bottou, et al. (2018)). In particular, one should have something like (in the paper's notation):
$$\mathbb{E}[F(w_T) - F^*] \leq (1 - \gamma)^T [F(w_0) - F^*] + \Delta.$$
This means that one can actually guarantee a fixed ratio of decrease in expectation to a neighborhood, whereas AdaScale converges linearly but not with a fixed ratio.

Some other small questions regarding the theory and experiments:
- Is there a reason why batch normalization was not tried for the CIFAR-10 experiments?
- Is it possible for $r_{t - 1} \gamma > 1$?
- Why was it necessary to estimate $\sigma^2 (w_t)$ and $\|\nabla F(w_t)\|^2$ by both aggregating at the current iteration and exponential averaging? What happens if exponential averaging is removed?
- What are the limitations of this method? How large of a batch size can one use with AdaScale before the algorithm breaks down (if at all)?

Additional Comments:

The algorithm is quite reminiscient of the steplength prescribed in Bollapragada, et al. (2018), which consider the steplength:
$$( 1 + \frac{\sigma^2(w_t)}{\|\nabla F(w_t)\|^2})^{-1}.$$
This gain ratio prescribed in this paper is the ratio between this quantity for two different batch sizes. Is there a clear explanation for why the relationship between these two quantities would arise?

This method could also be used for determining an appropriate scaling of the steplength in the sequential setting, when a larger batch size is used. Has this been considered?

Despite the concerns regarding the clarity in writing and the rigor in the theory of the paper, I think that the algorithmic idea proposed in this paper is interesting, novel, and practical. Because of the lack of clarity and rigor, I have given this paper a weak reject, but I would be happy to accept the paper if my concerns above were addressed in the final manuscript.

**Experience Assessment:**

I have published one or two papers in this area.

**Review Assessment: Checking Correctness Of Derivations And Theory:**

I carefully checked the derivations and theory.

**Review Assessment: Checking Correctness Of Experiments:**

I assessed the sensibility of the experiments.

**Review Assessment: Thoroughness In Paper Reading:**

I read the paper thoroughly.

---

> ### Author Response · Authors · 2019-11-14
> **Re: Official Blind Review #1**
>
> Thank you for the detailed and constructive review.
>
> We have updated all theorems to achieve the better rates.  Hopefully it is clear now that the theoretical comparisons are fair.
>
> We tried to write for multiple audiences, so that both deep learning practitioners and optimization experts would find the paper useful.  In Section 2, we define the scale S as the number of batches that are processed in parallel during each iteration.  Thus, batch size is proportional to scale.  We have updated the definition of “scale invariance” so that hopefully it is more clear.  AdaScale only approximately achieves scale invariance.  The scale invariant convergence bound (i.e., the bound that does not depend on S) formalizes this approximation, but the bound does not imply an exactly scale invariant algorithm.
>
> We have some experiments in Appendix C that show results with no exponential averaging.  There is more variance in the gain estimate, but AdaScale still performs well.
>
> In extreme settings, it is possible that r \gamma > 1.  In such cases, the bound would be Equation (6) in Appendix A of the updated submission.  In this case, AdaScale can oscillate between small and large r.  Related to this, there is indeed a limit on the scale S that achieves scale-invariant bounds, and we include this in the updated theorems.  In practice, however, we do not think r \gamma > 1 is a significant concern.  System and algorithm scaling inefficiencies limit the practicality of such extreme S.  Users can also decrease \gamma by decreasing the learning rate.
>
> Yes, one could apply AdaScale in the sequential setting, but we think AdaSale is most impactful for distributed training because of the large speed-ups.
>
> We do use batch normalization for CIFAR-10.  We do not train the batch norm scaling parameters because this helped us achieve 94% accuracy with 1 GPU.

---

### Official Review · AnonReviewer3 · 2019-10-23
**Official Blind Review #3**

**Rating:** 3

**Review:**

The paper presents and evaluate an algorithm, AdaScale SGD, to improve the stochastic gradient decent in distributed training. The proposed algorithm claims to be approximately scale-invariant, by adapting to the gradient's variance. The paper is well-written and generally easy to read, although I didn't check all theory in the paper.

The approach is evaluated using five benchmarks (networks / dataset combinations) from different  domains. The results are promising and seems solid.

The paper is good and I like it, although I think the novelty and contribution is slightly too low for ICLR. The kind of tweaking and minor optimizations to provide some adaptivity (or similar) in existing and established algorithms and approaches that is presented in this is paper is very important from a practical perspective. However, from a scientific perspective it provides no significant contribution.


**Experience Assessment:**

I have published one or two papers in this area.

**Review Assessment: Checking Correctness Of Derivations And Theory:**

I assessed the sensibility of the derivations and theory.

**Review Assessment: Checking Correctness Of Experiments:**

I assessed the sensibility of the experiments.

**Review Assessment: Thoroughness In Paper Reading:**

I made a quick assessment of this paper.

---

> ### Author Response · Authors · 2019-11-14
> **Re: Official Blind Review #3**
>
> We disagree that the novelty and contribution are too low for ICLR.  The ability to scale up training is often critical to the development of state-of-the-art models.  These models tend to be large, and developing them requires fast turnaround times and a lot of data.  AdaScale makes it significantly easier to scale up training, since the need to re-tune learning rate parameters is one of the biggest pain points when scaling to large batch sizes.  AdaScale is an entirely novel algorithm, as no prior algorithm adapts to the gradient’s variance in order to achieve this scaling goal.
>
> Furthermore, while we agree that AdaScale uses a tweak to provide adaptivity, we are surprised this is framed as a criticism.  Highly influential prior algorithms, such as Adam, AdaGrad, etc., could also be considered tweaks.  These tweaks are simple but non-trivial, and the simplicity has made these algorithms extremely successful.
>
> Finally, we do not understand the remark that “this is paper is very important from a practical perspective . . . from a scientific perspective it provides no significant contribution.”  The comment lacks justification and unfairly criticizes our paper’s importance.  Our paper introduces a practical training algorithm, thoroughly demonstrates its usefulness, and supports the algorithm with theoretical understanding.  These contributions fall firmly within ICLR’s established standards, and we believe many ICLR attendees would take interest in the work and find it useful.

---

### Official Review · AnonReviewer2 · 2019-10-26
**Official Blind Review #2**

**Rating:** 3

**Review:**

Authors use the PL condition to motivate a formula for learning rate selection. (they bound their loss by a quadratic with fixed curvature alpha, and use this to get learning rate).

Their analysis very closely mirrors one presented in "Gradient Diversity" paper, it uses the same assumptions on the loss. IE compare A.1 in the paper to the B.5 in "Gradient Diversity"

Gradient Diversity solves for batch size B after which linear learning rate scaling starts to break down, while this paper instead fixes B and solves for learning rate. Two results are comparable, if you take their learning rate formula in section 3.2 (need formula numbers) and solve for B which gives learning rate halfway between B=1 and B=inf, you get the same expression as "critical batch size" in equation 5 of gradient diversity paper.

It's not immediately obvious how to invert formula in Gradient Diversity paper to solve for learning rate, so I would consider their learning rate formula an interesting result.

I also appreciate the breadth of experiments used for evaluation.

The biggest issue I have with the paper is that I can't tell if it's better of worse than linear learning rate scaling from their experiment section. All of their experiments use more iterations for AS evaluation uses than for LSW evaluation. They demonstrate better training (and test) losses for AS, but because of extra iterations, I can't tell that the improvement in training loss is due to number of iterations, or due to AS scaling.

**Experience Assessment:**

I have read many papers in this area.

**Review Assessment: Checking Correctness Of Derivations And Theory:**

I carefully checked the derivations and theory.

**Review Assessment: Checking Correctness Of Experiments:**

I carefully checked the experiments.

**Review Assessment: Thoroughness In Paper Reading:**

I read the paper thoroughly.

---

> ### Author Response · Authors · 2019-11-14
> **Re: Official Blind Review #2**
>
> Thank you for the helpful review.
>
> Regarding the comparison with LSW:
>
> As suggested, we tested a version of LSW that uses more iterations.  We call this scaling strategy “LSW+”.  LSW+ scales the iterations axis of the LSW learning rate schedule so that LSW+ uses the same number of iterations as AdaScale.  The learning rate axis remains the same as LSW.
>
> The behavior of LSW+ is generally similar to that of LSW.  As expected, LSW+ improves upon LSW, but LSW+ still degrades model quality for all benchmarks at the largest scale (CIFAR, Deep Speech, ImageNet, Transformer, and YOLO).  For Deep Speech, Transformer, and YOLO, LSW+ also diverges at the same scales that LSW diverges.  We have included these results in updated Appendix D (Table 5).
>
> We also note that LSW+ is not a practical algorithm, because it requires either (i) first running AdaScale to determine the number of iterations; or (ii) tuning the number of iterations.  Both options are inconvenient in practice.  Moreover, for a fair comparison, it seems we would also need to consider AdaScale with tuning.  Thus, even if LSW+ had matched AdaScale, AdaScale would still be preferable to LSW+.
>
>
> Regarding the Gradient Diversity Paper:
>
> We agree our analysis shares similarities with Yin et al. (2018)’s analysis — several papers that we cite in Section 5 contain similar analysis.  But the contributions of the papers are different.  Yin et al. show that if we optimize the learning rate for convergence bounds, then large variance implies near-linear speed-ups from data parallelism, while small variance implies small speed-ups.
>
> What our paper shows is that regardless of how you set the learning rate schedule, our adaptive algorithm can achieve similar model quality at many different scales.  This algorithm is useful for many problems, regardless of whether the gradient’s variance is small, large, or constantly changing (as is common in practice).  In contrast, Yin et al. provide thresholds of the variance for which existing algorithms can work well or become less useful.
>
> Overall, we believe AdaScale is an important step toward user-friendly distributed training, and we think many ICLR attendees would find the algorithm interesting and useful.

---

### Public Comment · ~Michael_Petrochuk1 · 2019-09-28
**Excited but cautiously skeptical**

Thanks for this contribution to the community! Thanks for all the hard work.

This paper is exciting primarly because it claims that the AdaScale SGD optimizer has "no tuning parameters" and "often trains at a wide range of scales with nearly identical results". That said, I had a couple of questions:
- AdaScale includes a learning rate parameter. For the experimental results, what was the learning rate set at? Did tuning the learning rate help?
- In the experimental results, AdaScale produced nearly identical results. Was the experimental performance similar to the tuned state-of-the-art performance in each task? Does AdaScale trade performance for consistency?
- ALBERT was recently released and claimed state-of-the-art results across a number of NLP tasks. To accomplish that, it used the LAMB (updated LARS) optimizer. This paper claims that AdaScale is more general than LARS. Do you have any metrics comparing AdaScale to LAMB or LARS?
- Will you be releasing the code to replicate the experimental results?

Thanks again!

---

> ### Author Response · Authors · 2019-10-03
> **Hi Michael, thank you for your interest and helpful comments.**
>
> 1. AdaScale does not introduce new tuning parameters, but like momentum-SGD, we still must specify a learning rate schedule.  Hopefully it is clear from the text that we use only one simple learning rate schedule for each benchmark.  Between scales, we do not change these schedules.  Instead, AdaScale and LSW adapt the learning rate values, according to the definition of these strategies.
>
> The specific learning rate schedules are included in Table 3 (appendix).  For CIFAR, we tested an entire family of learning rate schedules (Figure 5).
>
> 2. Our benchmarks are quite standard, such as training ResNet-50 on ImageNet.  In all cases, the final model quality for single-batch training at minimum matches the reference implementation.
>
> 3. We do not have these comparisons.  We claim that AdaScale is more general than LARS because LARS is designed for CNNs.
>
> In many ways, these approaches are orthogonal to AdaScale, as they still rely on tuning or heuristics (such as root scaling and warm-up) to adapt to larger scales.  AdaScale is a principled and practical algorithm for adapting to larger scales.  Meanwhile, AdaScale does not consider layer-specific learning rates or Adam-style updates.  We are optimistic that extensions of AdaScale will incorporate such ideas in the future.
>
> 4. We are enthusiastic about ensuring the reproducibility of our results.  We probably cannot release the SyncSGD implementation we used for these experiments.  We more likely will release an open-source implementation of AdaScale for an existing open-source framework.

---

### Public Comment · ~Boris_Ginsburg1 · 2019-09-28
**Very cool and interesting work**

Congratulation on very cool and interesting work, and on well written paper!
A few questions:
1. Table 2. compares AdaScale (AS) with "Linear Scale" (LSW). LSW has significantly less iterations than AS. I wonder if you have LSW results when model  was trained for the same number of iterations as AS?
2. Speech-recognition (DeepSpeech2): you trained model only on train-clean part, and evaluated on test-clean. Probably this explains unusually high WER, and makes comparison with reference paper difficult. Is WER are for greedy decoder?  Do you have results for trainng on compete LibriSpeech (clean + other)?
3. Transformer: was it transformer-base or big? Can you add the description of what BLEU script have you used, please?

Comment:
p.5-  " To our knowledge, there are no standard schedules for solving speech and transformer with momentum-SGD".
The recipe for training DS2 with SGD with momentum:
https://nvidia.github.io/OpenSeq2Seq/html/speech-recognition/deepspeech2.html#training

---

> ### Author Response · Authors · 2019-10-03
> **Hi Boris, we appreciate your enthusiasm and helpful comments.**
>
> 1. We will work on this comparison. Please note that running LSW for the same number of iterations is significantly less practical than simply running AdaScale. It requires either (i) first running AdaScale to determine the number of iterations; or (ii) tuning the number of iterations.  The second path is inconvenient in practice (and furthermore, it seems we would also need to consider AdaScale with tuning).
>
> We emphasize that AdaScale achieving sublinear speedups is a feature, not a bug. SGD is not perfectly parallelizable.  It is important to have algorithms that account for this in a natural way.
>
> 2. Yes, we used the greedy decoder. We should note that consistency of model quality across scales is more important for this evaluation than the absolute model quality, since we are evaluating the approximate scale invariance of AdaScale.
>
> Our setup for the speech task closely follows the 'deep_speech' example from the TensorFlow repository.  This includes the data.  We only (i) changed the optimizer from Adam to momentum-SGD, and (ii) tuned initial/final learning rates and total steps (until our WER (~20%) matched that of the reference implementation (~23%)).  According to the reference implementation, our WER numbers are expected.
>
> Thus, we think that our speech benchmark is suitable for this evaluation.  We will also look into your suggestions for improving WER, since doing so would provide further data points on the performance of AdaScale.
>
> 3. The model is Transformer base (Table 1). We used the BLEU script from the TensorFlow repository. We will add a note about this in future versions.

---

### Author Response · Authors · 2019-11-14
**Updates to submission**

Our revised version makes the following changes:

* Results of stronger baseline included in Appendix D (Table 5)
* Updated Theorems with tighter bounds
* Updated gain ratio definition with expectations (importantly, this does not change estimation procedure, AdaScale implementation, experiments, etc.) -- it is only necessary for the theory section
* Updated definition of scale invariant algorithm

---

### Decision · Program_Chairs · 2019-12-19

**Decision:**

Reject

**Comment:**

Main summary: Novel rule for scaling learning rate, known as gain ration, for which the effective batch size is increased.

Discussion:
reviewer 2: main concern is reviewer can't tell if it's better of worse than linear learning rate scaling from their experiment section.
reviewer 3: novlty/contribution is a bit too low for ICLR.
reviewer 1: algorthmic clarity lacking.
Recommendation: all 3 reviewers recommend reject, I agree.